# Development and temporal validation of a nomogram for predicting ICU 28-day mortality in middle-aged and elderly sepsis patients: An eICU database study

Xiao She[1], Xiao Zhao[1], Haiyan Yang[1], Xiaoguang Cui[2]*

**1** Department of Gastroenterology, Xi'an Jiaotong University Second Affiliated Hospital, Xi'an, Shaanxi, China, **2** Department of Rheumatology and Immunology, Xi'an Jiaotong University Second Affiliated Hospital, Xi'an, Shaanxi, China

* cuixiaoguang@xjtu.edu.cn

## Abstract

### Background and objective

Despite advances in intensive care, sepsis remains a leading cause of mortality in intensive care unit (ICU) patients, especially middle-aged and elderly individuals. Given the limitations of conventional scoring systems and the interpretability challenges of machine learning models, this study aims to develop and temporally validate a nomogram for predicting 28-day ICU mortality in middle-aged and elderly sepsis patients via the eICU database (2014--2015), providing a clinically practical prediction tool.

### Methods

This retrospective study included 13,717 sepsis patients aged ≥45 years. The cohort was temporally divided into training (n = 6,397, 2014) and validation (n = 7,320, 2015) sets. Variable selection was performed via random forest importance ranking and LASSO regression. A nomogram was developed on the basis of multivariable logistic regression analysis.

### Results

The 28-day ICU mortality rates were 9.08% and 9.49% in the training and validation cohorts, respectively. The final nomogram incorporated 11 independent predictors: red cell distribution width (RDW), SOFA score, lactate, pH, 24-hour urine output, platelet count, total protein, temperature, heart rate, GCS score, and white blood cell (WBC) count. The model showed good discrimination in both the training (AUC: 0.805) and validation (AUC: 0.756) cohorts. The calibration curves demonstrated good agreement between the predicted and observed probabilities.

**Data availability statement:** The original source data are available at https://eicu-crd.mit.edu/. The specific dataset used for this study, including extracted variables and processed data for analysis, is available in the Dryad Digital Repository at https://doi.org/10.5061/dryad.hmgqnk9wb.

**Funding:** The author(s) received no specific funding for this work.

**Competing interests:** The authors have declared that no competing interests exist.

**Abbreviations:** ICU, Intensive Care Unit; SOFA, Sequential Organ Failure Assessment; SIRS, Systemic Inflammatory Response Syndrome; MODS, Multiple Organ Dysfunction Syndrome; APACHE, Acute Physiology and Chronic Health Evaluation; SAPS, Simplified Acute Physiology Score; ML, Machine Learning; GCS, Glasgow Coma Scale; RDW, Red Cell Distribution Width; WBC, White Blood Cell; PLT, Platelet; BMI, Body Mass Index; MAP, Mean Arterial Pressure; EPV, Events Per Variable; PPV, Positive Predictive Value; NPV, Negative Predictive Value; DCA, Decision Curve Analysis; AUC, Area Under the Curve.

## Conclusions

We developed and temporally validated a nomogram with good predictive performance for 28-day ICU mortality in middle-aged and elderly sepsis patients, providing a practical tool for risk stratification and clinical decision-making.

## Introduction

Sepsis is a common critical illness in the intensive care unit (ICU) and is characterized by systemic inflammatory response syndrome (SIRS) and multiple organ dysfunction syndrome (MODS) caused by infection [1]. Despite advances in diagnosis and organ support in intensive care units, sepsis continues to have high morbidity and mortality rates [2]. According to recent meta-analyses, the hospital mortality rate for sepsis patients is 26.7%, with an even higher rate of 41.9% among ICU patients [3]. In the ICU, mortality rates among sepsis patients are significantly higher than those in the general patient population [4]. Moreover, with an aging population, the number of elderly sepsis patients continues to increase, and these patients face increased mortality risk [5,6].

Early identification and accurate prediction of mortality risk in elderly sepsis patients are essential for improving clinical outcomes. While existing scoring systems provide valuable guidance in clinical practice, our nomogram model aims to offer more individualized risk assessment for middle-aged and elderly sepsis patients by integrating multiple clinical and laboratory parameters. This approach has the potential to complement existing tools, particularly in identifying high-risk patients within this specific population. Furthermore, these traditional scoring systems incorporate limited predictive variables, thus failing to comprehensively capture the complex pathophysiological processes underlying sepsis.

In recent years, machine learning (ML) models have been widely applied in the prognostic prediction of sepsis patients. For example, Liu et al. developed a stacking ensemble ML model based on the MIMIC-IV database to predict in-hospital mortality risk in patients with sepsis-induced coagulopathy. Their model identified anion gap and age as the most crucial predictive features, demonstrating robust predictive performance (AUC = 0.795, 95% CI: 0.763–0.827) [7]. Additionally, Wang et al. (2022) reported that the LightGBM model outperformed other ML algorithms in predicting 30-day mortality among sepsis patients, achieving an AUC of 0.90 [8]. Machine learning models demonstrate significant advantages in medical prediction by processing complex nonlinear relationships and integrating numerous clinical variables. However, the 'black box' nature of these models (the opacity of the decision-making process and the algorithmic complexity) can make it difficult for clinicians to understand and trust their predictions [9]. Furthermore, the model's generalizability may be limited by data heterogeneity and temporal variations, limiting its application across different healthcare settings.

With the accumulation of electronic health records and advancements in machine learning technologies, numerous studies have focused on developing models to

predict mortality risk in sepsis patients, aiming to provide robust support for clinical decision-making. While Shen et al. employed interpretable machine learning methods with multicenter validation, their study did not address validation differences across different years [10]. Yang et al. developed a conformity prediction model (CPMORS) incorporating model interpretation and uncertainty estimation; however, their validation data were confined to a specific time period, making it difficult to reflect the model's dynamic performance in clinical practice [11]. Although Zhang et al. developed an XGBoost model on the basis of multiple databases, their study also lacked stratified validation across different years [12]. The importance of temporal validation lies in the fact that diagnostic criteria, treatment protocols, and patient population characteristics may change over time in medical practice, potentially affecting the accuracy and reliability of prediction models. For example, in a gestational diabetes risk prediction study [13], the original model showed reasonable discrimination but suboptimal calibration during temporal validation, highlighting the importance of updating models to maintain their predictive performance in contemporary populations.

Middle-aged and elderly patients are more susceptible to sepsis and have poorer prognoses due to decreased physiological function, increased chronic disease burden, and impaired immune function. Therefore, developing and validating sepsis mortality prediction models specifically for this vulnerable population has significant clinical implications. However, current research on sepsis mortality prediction focuses primarily on model development for the general population, with relatively insufficient targeted studies for middle-aged and elderly patients as a high-risk group. For example, although Zhang et al. (2021) developed the Sepsis Mortality Risk Score (SMRS), their study population did not specifically distinguish middle-aged and elderly patients, and their dataset was limited to 2008--2012, failing to cover a broader time range to evaluate the model's long-term performance in this age group [14]. Other outcome prediction studies in sepsis patients also lack a specific focus on middle-aged and elderly patients [15,16].

Among medical prediction models, nomograms, as intuitive and practical visualization tools, have been widely applied in prognostic assessment and risk prediction for various diseases [17,18]. By transforming complex predictive models into visual graphics, nomograms enable clinicians to quickly and conveniently calculate patients' prognostic risk, demonstrating significant advantages in clinical practice. However, nomograms have not yet been developed for sepsis patients.

Given the limitations of existing predictive models, this study leverages the multicenter eICU database, which offers comprehensive and high-quality data particularly suitable for elderly sepsis patients. We aimed to develop a nomogram-based model using cross-year data (2014 for development, 2015 for validation) to predict 28-day ICU mortality in middle-aged and elderly sepsis patients. This novel approach, which incorporates temporal validation and nomogram visualization, addresses a significant gap in previous research. By comparing the validation results across different years, we can assess the model's stability and generalizability under evolving healthcare environments and treatment guidelines. This comprehensive evaluation provides valuable insights for clinical decision-making in this high-risk population and can also guide future model optimization, facilitating early risk stratification and personalized intervention strategies.

## Methods

### Data source and ethics approval

We utilized the eICU Collaborative Research Database (eICU-CRD), which contains clinical information from 335 intensive care units across 208 hospitals in the United States from 2014--2015. We accessed the database after completing the CITI program training and obtaining PhysioNet certification (Record ID: 67403327). The database is publicly available and fully deidentified in compliance with HIPAA regulations (Privacert, Cambridge, MA; Certification No. 1031219−2).

### Study population

The study enrolled patients who were diagnosed with sepsis upon admission to the intensive care unit (ICU). In accordance with the Sepsis-3 criteria, sepsis was defined as the presence of suspected or documented infection combined with an acute increase in the Sequential Organ Failure Assessment (SOFA) score of ≥2 points from baseline [19]. The

identification of infections was performed via the International Classification of Diseases, Ninth Revision (ICD-9) codes, and the physiological parameters necessary for SOFA score calculation were extracted from the Acute Physiology and Chronic Health Evaluation (APACHE) IV dataset [20]. According to the Sepsis-3 criteria, we included patients who were diagnosed with sepsis upon ICU admission. After excluding duplicate admissions, ICU stays less than 24 hours, patients younger than 45 years, and those with missing ICU outcome data, 13,717 patients were finally included and chronologically divided into training (2014, n = 6,397) and validation (2015, n = 7,320) cohorts.

## Data collection

Clinical data were collected from various tables within the eICU-CRD during the first 24 hours of ICU admission. Demographic characteristics (age, gender, ethnicity, body mass index [BMI]) and hospital admission data were extracted from the patient and apachePatientResult tables. The ApacheApsVar table provides vital signs (heart rate, respiratory rate, temperature, mean arterial pressure [MAP], and oxygen saturation), treatment-related variables (mechanical ventilation, dialysis, and vasopressor use), and severity scores (Glasgow Coma Scale [GCS], Sequential Organ Failure Assessment [SOFA], Acute Physiology and Chronic Health Evaluation IV [APACHE IV], and Acute Physiology Score III). Laboratory data, including blood gas parameters, complete blood count, blood chemistry, liver function tests, and coagulation profiles, were obtained from the laboratory table. Medical history was identified from the diagnosis table, while the site of infection was extracted from the AdmissionDx table.

## Outcome and sample size

Death within 28 days after ICU admission was considered the primary outcome event. In accordance with a previous study [21], the sample size was determined on the basis of the rule of at least 10 events per variable (EPV). A total of 13,717 patients were enrolled, with 581 deaths in the training cohort (n = 6,397) and 695 deaths in the validation cohort (n = 7,320). With 11 variables in the final model and 52.8 events per variable in the training cohort, our sample size substantially exceeded the minimum requirement determined by the EPV approach.

## Statistical analysis

The baseline demographic and clinical characteristics of all participants at admission are presented as the means (standard deviations) or medians (interquartile ranges) for continuous variables and as frequencies (percentages) for categorical variables, stratified by training and validation cohorts. Differences between groups were analyzed via the χ² test for categorical variables and one-way ANOVA or the Kruskal-Wallis test for normally and nonnormally distributed continuous variables, respectively.

From the univariate analysis, we identified variables significantly associated with 28-day ICU mortality ($p < 0.05$), removed those with high multicollinearity (VIF > 5), and further refined our selection through random forest importance ranking and LASSO regression with 10-fold cross-validation, ultimately identifying 11 key predictors for the final model (detailed in S1–S3 Tables and S1 Fig).

Owing to missing data for predictor variables, the final complete case analysis included 420 patients in the training cohort and 491 patients in the validation cohort. To assess potential selection bias, we compared baseline characteristics between patients with complete data (n = 911) and those with missing data (n = 12,806).

The final predictive model was constructed via multivariable logistic regression, incorporating the variables selected through both random forest importance ranking and LASSO regression. Model performance was assessed through the area under the receiver operating characteristic curve (AUC), sensitivity, specificity, positive predictive value (PPV), and negative predictive value (NPV). Calibration was evaluated via calibration curves, and clinical utility was assessed via decision curve analysis (DCA). The final model was visualized as a nomogram.

All the statistical analyses were two-tailed, with P<0.05 considered statistically significant. Analyses were performed via EmpowerStats (www.empowerstats.com, X&Y Solutions, Inc., Boston, MA) and R software version 4.2.0 (http://www.r-project.org).

## Results

The patient selection process is illustrated in Fig 1. From the eICU database (2014–2015), 23,136 patients with a diagnosis of sepsis on ICU admission were initially identified. After several exclusion steps, 13,717 patients were included in the study and divided into training and validation cohorts. After patients with missing predictor variables were excluded, 911 patients remained for the final analysis, comprising 420 patients in the training cohort (2014) and 491 patients in the validation cohort (2015). The 28-day ICU mortality rates were 9.08% and 9.49% in the training and validation cohorts, respectively.

To assess potential selection bias, we compared baseline characteristics between patients with complete data (n = 911) and those with missing data (n = 12,806). Patients with complete data demonstrated greater illness severity (SOFA score: 6.3±3.4 vs 4.2±2.8, P<0.001), whereas other baseline characteristics were largely comparable between the groups (S4 Table).

The baseline characteristics of both cohorts are presented in S5 Table. The training and validation cohorts presented similar demographic and clinical characteristics, with comparable disease severity, as measured by the SOFA and

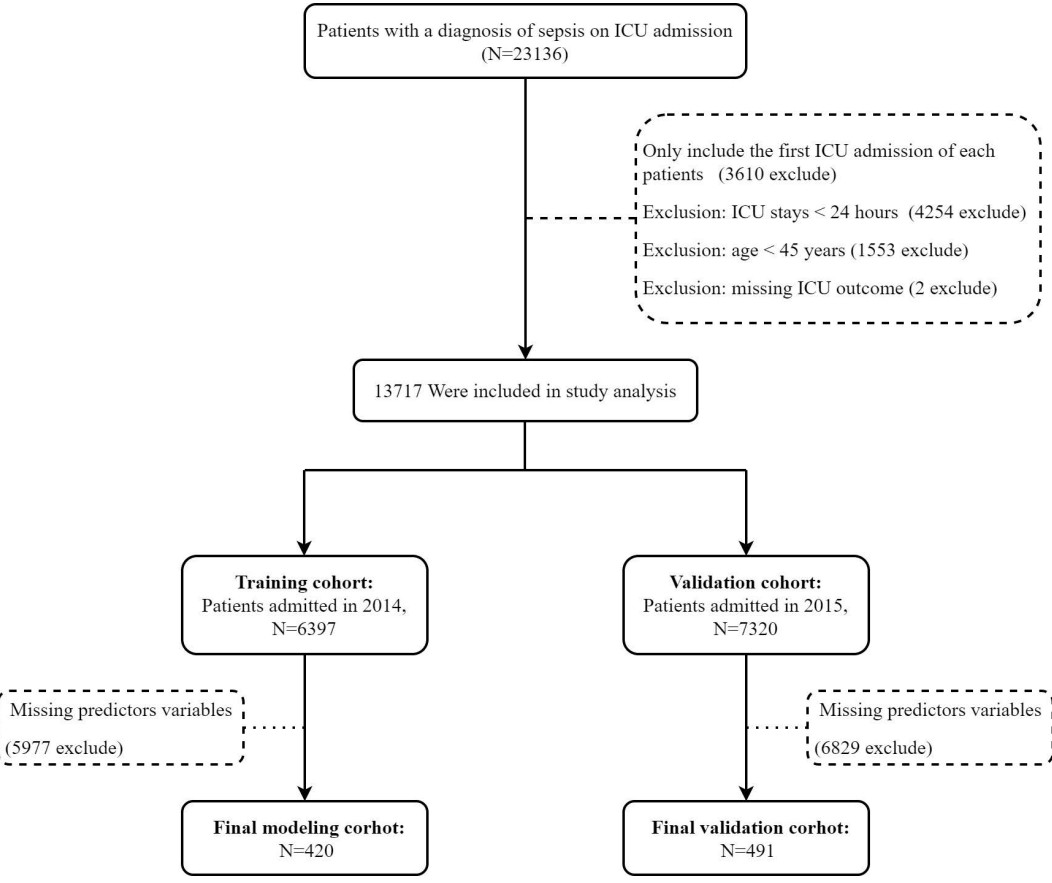

**Fig 1. Flow chart of the study.** *ICU* intensive care unit.

APACHE IV scores. Only minor differences were observed in respiratory parameters, lactate levels, and certain comorbidities between the cohorts.

The comparisons between survivors and nonsurvivors are presented in Table 1. In both cohorts, nonsurvivors were characterized by advanced age and lower BMI, while no significant differences were observed in sex distribution or ethnicity. Nonsurvivors were more frequently admitted through nonemergency departments. With respect to vital signs, nonsurvivors presented significantly higher heart and respiratory rates, accompanied by lower temperature and mean arterial pressure. The laboratory parameters revealed that nonsurvivors had more severe metabolic disturbances, enhanced inflammatory responses, compromised coagulation profiles, and impaired organ function. Disease severity scores (GCS, SOFA, APACHE IV, and APS III) were all significantly worse in nonsurvivors. With respect to comorbidities, nonsurvivors had higher incidences of congestive heart failure, acute myocardial infarction, and pneumonia but lower rates of diabetes. Furthermore, nonsurvivors require more intensive therapeutic interventions, including mechanical ventilation and vasopressor support.

Our model demonstrated slightly better discriminatory power (AUC: 0.813, 95% CI: 0.750–0.877) than the previously published SMRS scoring system [14] (AUC: 0.789, 95% CI: 0.720–0.859) for predicting sepsis mortality in our study population (Fig 2).

Our predictive model showed good discrimination in both the training cohort (AUC: 0.805) and the validation cohort (AUC: 0.756), with consistent sensitivity, specificity, and predictive values across cohorts (Table 2, Fig 3). The calibration curves confirmed good agreement between the predicted and observed mortality probabilities (Fig 4). Specifically, the development cohort showed good calibration for predicted probabilities less than 0.6, whereas the validation cohort exhibited better overall calibration performance. Decision curve analysis (DCA) revealed that the model provided greater net benefit than either the "treat all" or "treat none" strategies across threshold probabilities ranging from 0.1--0.8 in the development cohort and from 0.1--0.9 in the validation cohort (S2 Fig).

To facilitate clinical application, we developed a nomogram with 11 independent predictors (Fig 5): RDW, SOFA score, lactate, pH, 24-hour urine output, platelet count, total protein, temperature, heart rate, GCS score, and WBC count. For example, a patient with an RDW of 18% (35 points), a SOFA score of 11 (42 points), a lactate level of 8 mmol/L (28 points), a pH of 7.3 (22 points), a 24-h urine output of 1000 mL (58 points), a PLT (platelet) count of $150 \times 10^9$/L (32 points), a total protein level of 5 g/dL (25 points), a temperature of 38.5°C (18 points), a heart rate of 120 bpm (31 points), a GCS score of 11 (20 points), and a WBC count of $20 \times 10^9$/L (15 points) would total 326 points, corresponding to a predicted 28-day ICU mortality risk of 70%.

## Discussion

In this study, we analyzed 13,717 middle-aged and elderly sepsis patients from the eICU database (2014--2015). To address the limitations of existing prediction tools, we first evaluated the performance of previously published scoring systems in our cohort. Our analysis revealed that the current SMRS model had a moderate predictive ability (AUC: 0.789) for 28-day ICU mortality, suggesting room for improvement in risk stratification for this specific population.

Our research introduces innovation to the field of sepsis prognostic assessment through the implementation of a nomogram as a visual tool. Compared with traditional prediction methods, nomograms intuitively present the weights of various risk factors, enabling clinicians to more clearly understand each factor's influence on prognosis. We developed and validated a novel nomogram incorporating 11 clinical predictors that outperforms existing scoring systems, representing the first temporally validated risk stratification tool specifically designed for middle-aged and elderly sepsis patients and addressing a gap in existing assessment approaches regarding visual presentation and clinical practicality.

Recent studies have explored various machine learning approaches for sepsis mortality prediction. Zhang et al. [12] achieved remarkable performance when XGBoost (AUC: 0.94) was used to incorporate inflammatory biomarkers, whereas Bao et al. [16] reported excellent results when Light GBM was used (AUC: 0.99 training/0.96 testing). Our

**Table 1. Baseline clinical and laboratory characteristics of the study population stratified by 28-day ICU mortality.**

| | Training cohort (n=6397) | | | Validation cohort (n=7320) | | |
|---|---|---|---|---|---|---|
| | Survivors (n=5816) | Nonsurvivors (n=581) | P value | Survivors (n=6625) | Nonsurvivors (n=695) | P value |
| **Demographics** | | | | | | |
| Age (years) | 69.3±12.3 | 71.4±11.8 | <0.001 | 69.5±12.0 | 70.9±11.6 | 0.005 |
| BMI (kg/m$^2$) | 29.1±9.0 | 27.6±8.2 | <0.001 | 28.9±8.8 | 28.1±9.1 | 0.016 |
| Gender | | | 0.391 | | | 0.597 |
| Male | 2880 (49.5%) | 277 (47.7%) | | 3205 (48.4%) | 329 (47.3%) | |
| Female | 2933 (50.5%) | 304 (52.3%) | | 3418 (51.6%) | 366 (52.7%) | |
| Ethnicity | | | 0.177 | | | 0.864 |
| Caucasian | 4498 (77.3%) | 435 (74.9%) | | 5299 (80.0%) | 554 (79.7%) | |
| Other | 1318 (22.7%) | 146 (25.1%) | | 1326 (20.0%) | 141 (20.3%) | |
| Hospital admit source | | | <0.001 | | | 0.001 |
| Emergency Department | 2910 (50.0%) | 249 (42.9%) | | 3378 (51.0%) | 309 (44.5%) | |
| Other | 2906 (50.0%) | 332 (57.1%) | | 3247 (49.0%) | 386 (55.5%) | |
| **Vital signs** | | | | | | |
| Heart rate (/min) | 110.1±27.8 | 118.0±29.9 | <0.001 | 109.9±29.3 | 119.3±29.7 | <0.001 |
| Respiratory rate (bpm) | 29.0±14.4 | 32.5±13.5 | <0.001 | 30.1±14.2 | 33.9±13.4 | <0.001 |
| Temperature (℃) | 36.6±1.2 | 36.2±1.5 | <0.001 | 36.6±1.1 | 36.4±1.4 | <0.001 |
| MAP (mmHg) | 56.0 (47.0-110.0) | 52.0 (43.0-70.0) | <0.001 | 57.0 (48.0-112.0) | 50.0 (43.0-65.8) | <0.001 |
| O$_2$ Sat (%) | 93.6±9.2 | 93.3±9.6 | 0.532 | 94.5±7.9 | 93.3±9.0 | 0.007 |
| **Laboratory data** | | | | | | |
| PH | 7.4±0.1 | 7.3±0.1 | <0.001 | 7.4±0.1 | 7.3±0.1 | <0.001 |
| PaO$_2$ (mmHg) | 90.0 (70.0-130.0) | 93.0 (71.0-142.5) | 0.159 | 91.6 (73.0-132.0) | 89.0 (67.5-131.5) | 0.093 |
| PaCO$_2$ (mmHg) | 41.1±14.6 | 39.4±14.8 | 0.028 | 41.4±14.7 | 39.7±15.4 | 0.021 |
| FiO$_2$ (%) | 40.0 (29.0-70.0) | 62.5 (40.0-100.0) | <0.001 | 40.0 (29.0-70.0) | 60.0 (40.0-100.0) | <0.001 |
| Urine output (24 h, mL) | 1253.8 (638.9-2196.2) | 626.9 (194.0-1247.7) | <0.001 | 1303.4 (644.8-2280.5) | 617.7 (181.3-1282.3) | <0.001 |
| Lactate (mmol/L) | 1.7 (1.1-2.6) | 2.9 (1.7-4.9) | <0.001 | 1.8 (1.2-2.8) | 3.1 (1.7-5.6) | <0.001 |
| Bicarbonate (mmol/L) | 22.7±5.4 | 20.6±6.5 | <0.001 | 22.5±5.4 | 20.2±6.3 | <0.001 |
| Base Excess (mmol/L) | −1.6 (−6.5-2.1) | −5.7 (−11.5-1.0) | <0.001 | −2.0 (−6.8-2.0) | −6.0 (−11.1--0.8) | <0.001 |
| WBC count (cells x 10$^9$/L) | 13.2 (9.0-19.1) | 15.1 (9.0-22.2) | <0.001 | 13.0 (8.7-18.9) | 15.2 (9.1-21.9) | <0.001 |
| Hemoglobin (g/dL) | 10.4±2.0 | 10.3±2.2 | 0.793 | 10.4±2.1 | 10.1±2.2 | 0.002 |
| Platelets (cells x 10$^9$/L) | 183.0 (126.0-257.0) | 164.0 (93.0-247.5) | <0.001 | 183.0 (128.0-254.0) | 171.0 (94.5-248.5) | <0.001 |
| RDW (%) | 16.0±2.5 | 17.0±3.0 | <0.001 | 16.0±2.6 | 17.1±3.2 | <0.001 |
| MCHC (g/dL) | 32.7±1.5 | 32.6±1.6 | 0.377 | 32.5±1.5 | 32.3±1.6 | <0.001 |
| Albumin (g/dL) | 2.5±0.6 | 2.2±0.6 | <0.001 | 2.5±0.6 | 2.3±0.6 | <0.001 |
| Total protein (g/dL) | 5.7±0.9 | 5.3±1.1 | <0.001 | 5.8±0.9 | 5.4±1.0 | <0.001 |
| Glucose (mg/dl) | 128.0 (102.0-169.0) | 131.0 (99.0-171.0) | 0.739 | 129.0 (103.0-171.0) | 128.0 (98.0-174.0) | 0.220 |
| Sodium (mmol/L) | 138.3±6.3 | 138.4±7.1 | 0.739 | 138.3±6.2 | 138.7±6.9 | 0.212 |
| Serum potassium (mmol/L) | 4.1±0.8 | 4.2±0.9 | <0.001 | 4.1±0.7 | 4.3±0.9 | <0.001 |
| Calcium (mg/dl) | 8.0±0.8 | 7.8±1.0 | <0.001 | 8.0±0.8 | 7.9±1.1 | <0.001 |
| Serum creatinine (mg/dL) | 1.3 (0.9-2.3) | 1.9 (1.1-3.0) | <0.001 | 1.9±1.8 | 2.4±1.8 | <0.001 |
| BUN (mg/dL) | 29.0 (18.0-46.0) | 40.0 (25.0-58.5) | <0.001 | 28.0 (18.0-45.0) | 38.0 (26.0-57.0) | <0.001 |
| ALT (U/L) | 27.0 (17.0-52.0) | 35.0 (19.0-78.0) | <0.001 | 26.0 (16.0-51.0) | 35.0 (21.0-92.5) | <0.001 |
| AST (U/L) | 34.0 (20.0-70.0) | 58.0 (28.0-146.0) | <0.001 | 34.0 (20.0-70.0) | 63.5 (33.0-182.2) | <0.001 |
| Total bilirubin (mg/dL) | 1.2±1.9 | 2.4±4.3 | <0.001 | 0.7 (0.4-1.2) | 1.0 (0.6-2.1) | <0.001 |
| Anion gap (mmol/L) | 11.9±4.7 | 13.9±5.7 | <0.001 | 11.8±4.8 | 13.9±6.1 | <0.001 |
| PT (seconds) | 17.0 (14.7-22.2) | 19.1 (16.0-27.9) | <0.001 | 17.1 (14.7-22.8) | 21.0 (16.7-29.9) | <0.001 |

*(Continued)*

**Table 1.** (Continued)

| | Training cohort (n = 6397) | | | Validation cohort (n = 7320) | | |
|---|---|---|---|---|---|---|
| | Survivors (n = 5816) | Nonsurvivors (n = 581) | P value | Survivors (n = 6625) | Nonsurvivors (n = 695) | P value |
| APTT (seconds) | 40.1 ± 19.5 | 46.1 ± 22.8 | <0.001 | 35.0 (30.7-42.2) | 39.0 (33.3-50.0) | <0.001 |
| INR | 1.4 (1.2-2.0) | 1.7 (1.4-2.5) | <0.001 | 1.4 (1.2-2.1) | 1.8 (1.4-2.8) | <0.001 |
| **Site of infection** | | | <0.001 | | | <0.001 |
| Pulmonary | 2198 (37.8%) | 268 (46.1%) | | 2586 (39.0%) | 321 (46.2%) | |
| Other | 3618 (62.2%) | 313 (53.9%) | | 4039 (61.0%) | 374 (53.8%) | |
| **Severity of illness** | | | | | | |
| GCS score | 12.7 ± 3.2 | 10.8 ± 4.2 | <0.001 | 12.8 ± 3.2 | 11.2 ± 4.1 | <0.001 |
| SOFA score | 4.0 (2.0-6.0) | 6.0 (4.0-9.0) | <0.001 | 4.0 (2.0-6.0) | 7.0 (4.0-9.0) | <0.001 |
| Apache IV score | 69.5 ± 23.0 | 95.6 ± 29.8 | <0.001 | 68.5 ± 23.2 | 94.6 ± 29.9 | <0.001 |
| Acute Physiology Score III | 54.5 ± 22.0 | 79.2 ± 29.7 | <0.001 | 53.6 ± 22.0 | 77.7 ± 29.6 | <0.001 |
| **Past medical history** | | | | | | |
| COPD | | | 0.166 | | | 0.626 |
| No | 5220 (89.8%) | 532 (91.6%) | | 6016 (90.8%) | 635 (91.4%) | |
| Yes | 596 (10.2%) | 49 (8.4%) | | 609 (9.2%) | 60 (8.6%) | |
| CHF | | | 0.012 | | | 0.450 |
| No | 5290 (91.0%) | 510 (87.8%) | | 6043 (91.2%) | 628 (90.4%) | |
| Yes | 526 (9.0%) | 71 (12.2%) | | 582 (8.8%) | 67 (9.6%) | |
| AMI | | | 0.003 | | | 0.544 |
| No | 5654 (97.2%) | 552 (95.0%) | | 6361 (96.0%) | 664 (95.5%) | |
| Yes | 162 (2.8%) | 29 (5.0%) | | 264 (4.0%) | 31 (4.5%) | |
| DM | | | 0.013 | | | 0.042 |
| No | 4888 (84.0%) | 511 (88.0%) | | 5728 (86.5%) | 620 (89.2%) | |
| Yes | 928 (16.0%) | 70 (12.0%) | | 897 (13.5%) | 75 (10.8%) | |
| Pneumonia | | | <0.001 | | | 0.002 |
| No | 3817 (65.6%) | 329 (56.6%) | | 4440 (67.0%) | 426 (61.3%) | |
| Yes | 1999 (34.4%) | 252 (43.4%) | | 2185 (33.0%) | 269 (38.7%) | |
| Rhythm | | | <0.001 | | | <0.001 |
| No | 4818 (82.8%) | 423 (72.8%) | | 5562 (84.0%) | 541 (77.8%) | |
| Yes | 998 (17.2%) | 158 (27.2%) | | 1063 (16.0%) | 154 (22.2%) | |
| **Intervention** | | | | | | |
| Mechanical ventilation | | | <0.001 | | | <0.001 |
| No | 4180 (73.1%) | 295 (51.0%) | | 4754 (72.9%) | 379 (54.8%) | |
| Yes | 1537 (26.9%) | 283 (49.0%) | | 1763 (27.1%) | 313 (45.2%) | |
| Dialysis | | | 0.991 | | | 0.127 |
| No | 5411 (94.6%) | 547 (94.6%) | | 6182 (94.9%) | 647 (93.5%) | |
| Yes | 306 (5.4%) | 31 (5.4%) | | 335 (5.1%) | 45 (6.5%) | |
| Vasopressor use (1st 24 h) | | | 0.080 | | | 0.007 |
| No | 5655 (99.2%) | 564 (98.4%) | | 6471 (99.5%) | 678 (98.7%) | |
| Yes | 48 (0.8%) | 9 (1.6%) | | 32 (0.5%) | 9 (1.3%) | |

Data are expressed as the mean±SD, median (interquartile range), or percentage. *BMI*: Body mass index; *MAP*: Mean arterial pressure; *O₂ Sat*: Oxygen saturation; *PaO₂*: Partial pressure of arterial oxygen; *PaCO₂*: Partial pressure of arterial carbon dioxide; *FiO₂*: Fraction of inspired oxygen; *WBC*: White blood cell; *RDW*: Red cell distribution width; *MCHC*: Mean corpuscular hemoglobin concentration; *BUN*: Blood urea nitrogen; *ALT*: Alanine aminotransferase; *AST*: Aspartate aminotransferase; *PT*: Prothrombin time; *APTT*: Activated partial thromboplastin time; *INR*: International normalized ratio; *GCS*: Glasgow coma scale; *SOFA*: Sequential organ failure assessment; *APACHE*: Acute physiology and chronic health evaluation; *COPD*: Chronic obstructive pulmonary disease; *CHF*: Congestive heart failure; *AMI*: Acute myocardial infarction; *DM*: Diabetes mellitus.

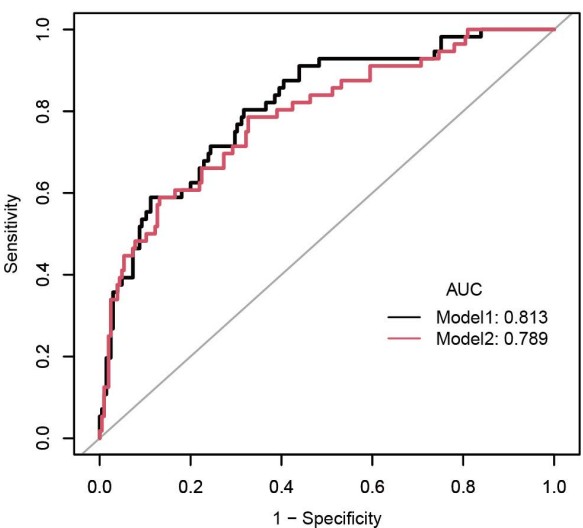

**Compare 2 models**

**Fig 2. Receiver operating characteristic (ROC) curves comparing our predictive model (Model 1, AUC = 0.813) with the SMRS model (Model 2, AUC = 0.789) for sepsis mortality.** *SMRS* Sepsis Mortality Risk Score.

**Table 2. Accuracy of the predictive model in the training and validation cohorts.**

| Model characteristics | Training cohort (n = 420) | Validation cohort (n = 491) |
|---|---|---|
| AUC (95% CI) | 0.805 (0.749-0.862) | 0.756 (0.700-0.812) |
| Threshold | −1.396 | −1.408 |
| Sensitivity, % | 73.81 | 70.59 |
| Specificity, % | 74.70 | 71.47 |
| Accuracy, % | 74.52 | 71.28 |
| Positive predictive value, % | 42.18 | 39.34 |
| Negative predictive value, % | 91.94 | 90.26 |

*AUC*: area under the curve; *CI*: confidence interval.

relatively lower AUC may reflect our focus on interpretability and clinical practicality rather than pure predictive performance. The use of readily available clinical predictors makes our model more accessible for routine clinical implementation, particularly in resource-limited settings.

Notably, our findings align with those of previous studies regarding key mortality predictors. The importance of APACHE scores, lactate levels, and organ dysfunction markers has been consistently demonstrated across studies. Yang et al. [11] identified similar risk factors in their CPMORS model (AUC: 0.858 internal/0.800 external), although they emphasized the value of uncertainty quantification through conformal prediction. Zhang et al. [12] further validated these findings via XGBoost (AUC: 0.94), identifying age, AST, invasive ventilation and BUN as the strongest predictors. Our nomogram approach offers comparable performance while maintaining transparency and ease of use.

Our study identified 11 key predictors of sepsis mortality, with RDW, the SOFA score, and the lactate level emerging as the most clinically significant. These three variables reflect critical aspects of sepsis pathophysiology, including the inflammatory status, organ dysfunction, and tissue hypoperfusion.

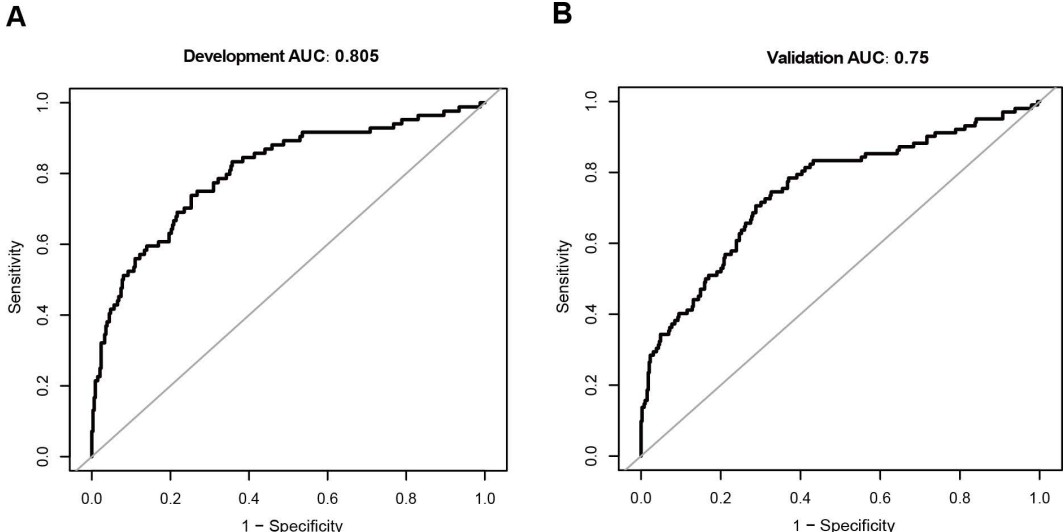

**Fig 3. ROC curves for predicting 28-day ICU mortality in middle-aged and elderly patients with sepsis.** (A) Development cohort (AUC = 0.805). (B) Validation cohort (AUC = 0.75).

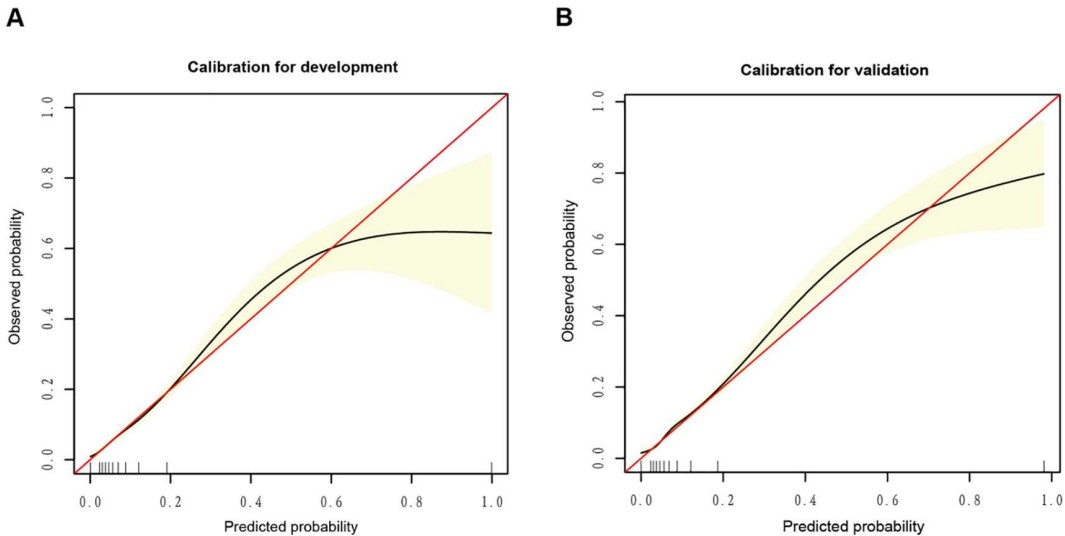

**Fig 4. Calibration curves for predicting 28-day ICU mortality in middle-aged and elderly patients with sepsis.** (A) Development cohort. (B) Validation cohort. The red diagonal line represents perfect calibration, the black curve represents the actual calibration, and the yellow shading indicates the 95% confidence interval.

These findings align with those of previous studies, further substantiating the selection of variables employed in our model and emphasizing the importance of multisystem evaluation in the prognosis of sepsis. Yang et al. [11] demonstrated that RDW and SOFA scores were significant predictors in their machine learning model. Specifically, elevated RDWs reflect increased oxidative stress and inflammatory cytokine production under chronic inflammatory conditions. These factors interfere with the intracellular metabolism of iron and vitamin B12, ultimately impairing erythropoiesis and resulting in increased RDWs [22]. Compared with existing sepsis prediction tools, our model specifically incorporates the RDW as

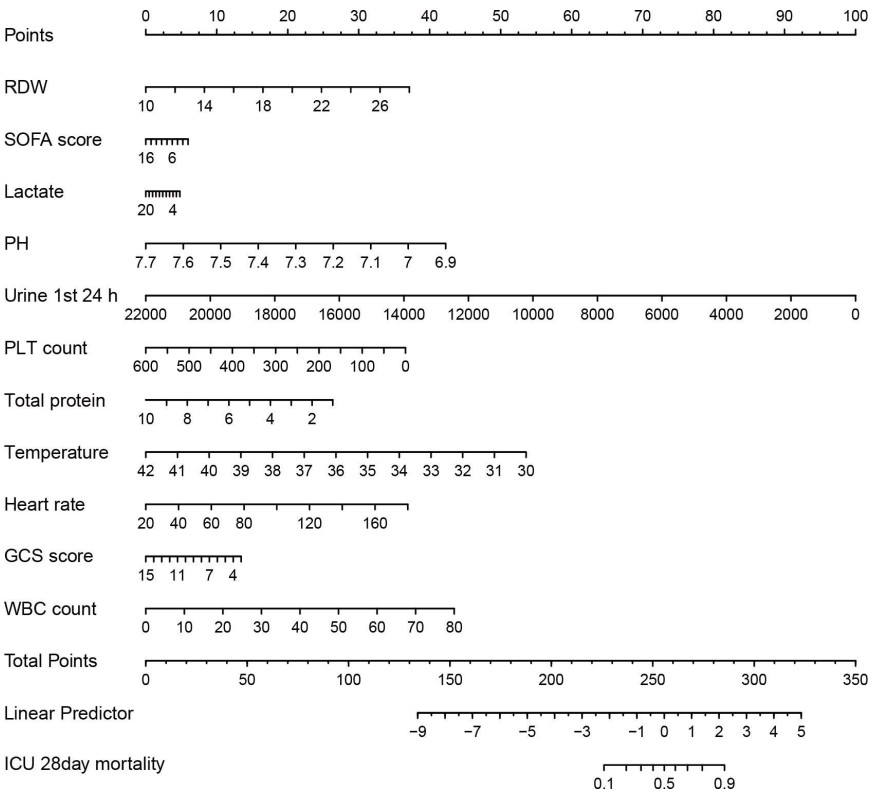

**Fig 5. Nomogram for predicting 28-day ICU mortality in middle-aged and elderly patients with sepsis.** To use this nomogram, firs the patient's value for each variable (RDW, SOFA score, lactate, pH, urine output in the first 24 h, PLT count, total protein, temperature, heart rate, GCS score, and WBC count) was located on the corresponding axis. A vertical line is drawn up to the "Points" axis to determine the points assigned for each variable. All points are summed to obtain the "Total Points". A vertical line is drawn from the "Total Points" axis down through the "Linear Predictor" line to the "ICU 28-day mortality" line to determine the predicted probability of 28-day ICU mortality.

a key predictor, which is relatively uncommon in traditional scoring systems. Despite the confirmed prognostic value of RDW, it remains underutilized in routine clinical sepsis scoring tools. By integrating RDW into our nomogram, we have expanded the biological foundation of prediction and provided an assessment method that better captures the effects of inflammatory status and oxidative stress. Bao et al. [16] confirmed that GCS scores and lactate levels are crucial prognostic indicators. Lower GCS scores suggest neurological dysfunction [23] and potential cerebral hypoperfusion [24], whereas elevated lactate levels indicate tissue hypoxia at the cellular level and the activation of anaerobic metabolism [25]. Similarly, Zhang et al. [12] identified platelet count and total protein levels as independent factors associated with mortality risk. Thrombocytopenia is often indicative of disseminated intravascular coagulation (DIC) [26] and endothelial dysfunction [27], whereas decreased total protein levels suggest capillary leak syndrome [28] and impaired protein synthesis due to organ dysfunction. Collectively, these observations reinforce the validity of the variables included in our model and highlight the necessity for comprehensive, multisystem evaluation in predicting outcomes in sepsis patients.

Other predictors in our model, while showing relatively lower importance, still contribute significantly to mortality prediction. The 24-hour urine output, as noted by Yang et al. [11], reflects both renal function and tissue perfusion. Oliguria may indicate acute kidney injury, decreased cardiac output, or increased vascular permeability, which are characteristic of septic shock [29–31]. The temperature and WBC count, although traditional markers, remain valuable in assessing the systemic inflammatory response. Fever represents enhanced immune activation and increased

metabolic demands [32], whereas abnormal white blood cell counts reflect the severity of the inflammatory response and immune status. Heart rate and pH provide important information about cardiovascular status and metabolic derangement, which Shen et al. [10] identified as critical components in their prediction model. Specifically, tachycardia reflects compensatory mechanisms to maintain cardiac output and the direct effects of inflammatory mediators on the myocardium [33,34]. Acidosis indicates severe cellular dysfunction and may further compromise cardiovascular function through reduced myocardial contractility [35] and impaired myocardial energy metabolism [36]. These parameters, although individually may appear less prominent, collectively enhance the model's ability to capture the complex pathophysiological alterations in sepsis.

Our nomogram provides clinicians with an intuitive tool to assess 28-day ICU mortality risk in middle-aged and elderly sepsis patients, potentially facilitating early identification of high-risk individuals and optimizing resource allocation in ICU settings." This not only clarifies the clinical significance of this study, but also avoids duplication of the results.

### Strengths and innovation

Our study has several key strengths: (1) temporal validation design using separate years (2014--2015) to assess model stability; (2) integration of readily available clinical parameters, making it practical for routine use; and (3) development of an intuitive visual nomogram that balances predictive accuracy with clinical interpretability, allowing clinicians to directly quantify the contribution of each risk factor without complex calculations. The nomogram enhances bedside application efficiency and makes the tool suitable for various healthcare settings, including resource-limited environments, providing a valuable complement to existing scoring systems; (4) comprehensive performance evaluation, including discrimination, calibration, and decision curve analysis; and (5) a specific focus on middle-aged and elderly sepsis patients, a demographic group with unique physiological challenges and increased mortality risks that may not be adequately captured by existing general tools. By targeting this critical population, our model more accurately assesses age-related risk factors and provides clinicians with a more tailored decision support tool.

### Limitations and future directions

Despite the use of real-world clinical data, which enhances the authenticity of our findings, several limitations should be noted in this study. First, the temporal validation period (2014--2015) is relatively short due to database constraints, warranting further validation across longer time spans. Second, substantial amounts of data were missing, with only 911 of the 13,717 patients having complete predictor variables. To address this limitation and assess potential bias, we compared baseline characteristics between patients with and without complete data. Although patients with complete data had greater illness severity (SOFA score: $6.3 \pm 3.4$ vs $4.2 \pm 2.8$, $P < 0.001$), most other clinical characteristics remained comparable between the groups. These findings suggest that our findings may be more applicable to critically ill sepsis patients, and caution should be exercised when these results are generalized to patients with milder disease severity. Third, external validation in different geographical populations is needed to confirm generalizability. Fourth, the retrospective nature of the study limits causal inference. Additionally, dynamic prediction incorporating temporal changes in clinical parameters could improve accuracy. Future studies should focus on prospective validation, integration with electronic health records, and evaluation of the tool's impact on clinical outcomes and decision-making processes.

### Conclusions

In conclusion, we developed and temporally validated a practical nomogram incorporating 11 readily available clinical parameters to predict 28-day ICU mortality in middle-aged and elderly sepsis patients. The model demonstrated good discriminative ability in both the training (AUC: 0.805) and validation (AUC: 0.756) cohorts, with consistent calibration and favorable decision curve analysis results. This nomogram provides clinicians with an intuitive tool for rapid risk stratification, potentially facilitating early identification of high-risk patients and supporting clinical decision-making. While

promising, further external validation in different populations and evaluation of the tool's clinical utility in real-world settings are warranted.

## Supporting information

**S1 Table. Univariate logistic regression analysis of 28-day ICU mortality in the training cohort.** Data are OR (95% CI) and P value. *BMI*: Body mass index; *MAP*: Mean arterial pressure; $O_2$ *Sat*: Oxygen saturation; $PaO_2$: Partial pressure of arterial oxygen; $PaCO_2$: Partialpressure of arterial carbon dioxide; $FiO_2$: Fraction of inspired oxygen; *WBC*: White blood cell; *RDW*: Red cell distribution width; *MCHC*: Mean corpuscular hemoglobin concentration; *BUN*: Blood urea nitrogen; *ALT*: Alanine aminotransferase; *AST*: Aspartate aminotransferase; *PT*: Prothrombin time; *APTT*: Activated partial thromboplastin time; *INR*: International normalized ratio; *GCS*: Glasgow coma scale; *SOFA*: Sequential organ failure assessment; *APACHE*: Acute physiology and chronic health evaluation; *COPD*: Chronic obstructive pulmonary disease; *CHF*: Congestive heart failure; *AMI*: Acute myocardial infarction; *DM*: Diabetes mellitus.
(DOCX)

**S2 Table. Multiple collinearity screening of variables in the training cohort.** Data are variance inflation factor (VIF) values. NA indicates the variable was removed from the model at that step. Variables eliminated (VIF > 5): Bicarbonate (VIF = 17.0), Base Excess (VIF = 18.2), ALT (VIF = 9.9), PT (VIF = 66.2), APACHE IV score (VIF = 69.6), and Acute Physiology Score III (VIF = 73.0). *BMI*: Body mass index; *GCS*: Glasgow coma scale; $PaCO_2$: Partial pressure of arterial carbon dioxide; $FiO_2$: Fraction of inspired oxygen; *WBC*: White blood cell; *RDW*: Red cell distribution width; *BUN*: Blood urea nitrogen; *ALT*: Alanine aminotransferase; *AST*: Aspartate aminotransferase; *PT*: Prothrombin time; *APTT*: Activated partial thromboplastin time; *INR*: International normalized ratio; *SOFA*: Sequential organ failure assessment; *APACHE*: Acute physiology and chronic health evaluation; *CHF*: Congestive heart failure; *AMI*: Acute myocardial infarction; *DM*: Diabetes mellitus.
(DOCX)

**S3 Table. Variable importance for 28-day ICU mortality prediction by random forest in the training cohort.** SET params as: n_trees = 500, split_features = 6, total_features = 34, sampling = swor (without replacement), resample_size = 4043, splitting_rule = gini random, random_split_points = 10, class_imbalance_ratio = 10.01. The table shows all important variables ranked by total importance score in the random forest model. Total importance represents the overall predictive power of each variable. Positive impact indicates the variable's contribution to predicting mortality when its value increases, while negative impact indicates its contribution to predicting survival. Analysis was performed using the training cohort (patients discharged in 2014, n = 6,397). *SOFA*: Sequential organ failure assessment; *WBC*: White blood cell; *RDW*: Red cell distribution width; $PaCO_2$: Partial pressure of arterial carbon dioxide; *GCS*: Glasgow coma scale; $FiO_2$: Fraction of inspired oxygen; *BUN*: Blood urea nitrogen; *INR*: International normalized ratio; *AST*: Aspartate aminotransferase; *APTT*: Activated partial thromboplastin time; *BMI*: Body mass index; *AMI*: Acute myocardial infarction; *CHF*: Congestive heart failure; *DM*: Diabetes mellitus.
(DOCX)

**S1 Fig. (A) Tenfold cross-validation plot showing the binomial deviance against log(lambda).** (B) Coefficient path plot showing variable selection with log(lambda) values.
(PNG)

**S4 Table. Comparison of baseline characteristics between patients with complete data and those with missing data.** Data are expressed as the mean±SD, median (interquartile range), or percentage. *BMI*: Body mass index; *MAP*: Mean arterial pressure; *O2 Sat*: Oxygen saturation; $PaO_2$: Partial pressure of arterial oxygen; $PaCO_2$: Partial pressure of arterial carbon dioxide; $FiO_2$: Fraction of inspired oxygen; *WBC* White blood cell; *RDW*: Red cell distribution width; *MCHC*: Mean

corpuscular hemoglobin concentration; *BUN*: Blood urea nitrogen; *ALT*: Alanine aminotransferase; *AST*: Aspartate amino-transferase; *PT*: Prothrombin time; *APTT*: Activated partial thromboplastin time; *INR*: International normalized ratio; *GCS*: Glasgow coma scale; *SOFA*: Sequential organ failure assessment; *APACHE*: Acute physiology and chronic health evaluation; *COPD*: Chronic obstructive pulmonary disease; *CHF*: Congestive heart failure; *AMI*: Acute myocardial infarction; *DM*: Diabetes mellitus.
(DOCX)

**S5 Table. Baseline clinical and laboratory characteristics of the study population in the training and validation cohorts.** Data are expressed as the mean±SD, median (interquartile range), or percentage. *BMI*: Body mass index; *MAP*: Mean arterial pressure; *O2 Sat*: Oxygen saturation; *PaO$_2$*: Partial pressure of arterial oxygen; *PaCO$_2$*: Partial pressure of arterial carbon dioxide; *FiO$_2$*: Fraction of inspired oxygen; *WBC* White blood cell; *RDW*: Red cell distribution width; *MCHC*: Mean corpuscular hemoglobin concentration; *BUN*: Blood urea nitrogen; *ALT*: Alanine aminotransferase; *AST*: Aspartate aminotransferase; *PT*: Prothrombin time; *APTT*: Activated partial thromboplastin time; *INR*: International normalized ratio; *GCS*: Glasgow coma scale; *SOFA*: Sequential organ failure assessment; *APACHE*: Acute physiology and chronic health evaluation; *COPD*: Chronic obstructive pulmonary disease; *CHF*: Congestive heart failure; *AMI*: Acute myocardial infarction; *DM*: Diabetes mellitus.
(DOCX)

**S2 Fig. Decision curve analysis for predicting 28-day ICU mortality in middle-aged and elderly patients with sepsis.** (A) Development cohort. (B) Validation cohort. The grey line represents the net benefit of treating all patients, the black line represents treating no patients, and the red line represents the net benefit of the prediction model at different threshold probabilities. *ICU*, Intensive Care Unit; *DCA*, Decision Curve Analysis.
(PNG)

## Acknowledgments

We sincerely thank the eICU-CRD for providing valuable data that significantly contributed to our study.

## Author contributions

**Conceptualization:** Xiao She, Haiyan Yang, Xiaoguang Cui.

**Data curation:** Xiao Zhao, Haiyan Yang.

**Formal analysis:** Xiao She, Xiao Zhao, Haiyan Yang.

**Methodology:** Xiao She, Xiao Zhao, Xiaoguang Cui.

**Writing – original draft:** Xiao She.

**Writing – review & editing:** Xiaoguang Cui.

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
