## [Decision Letter · Decision Letter 0]

Dear Dr. Cui,

We look forward to receiving your revised manuscript.

Kind regards,

Robert Jeenchen Chen, MD, MPH, ChFC®, EA

Academic Editor

PLOS ONE

Journal Requirements:

*2**.* Thank you for uploading your study's underlying data set. Unfortunately, the repository you have noted in your Data Availability statement does not qualify as an acceptable data repository according to PLOS's standards.

Reviewers' comments:

Reviewer's Responses to Questions

**Comments to the Author**

1. Is the manuscript technically sound, and do the data support the conclusions?

Reviewer #1: Yes

Reviewer #2: Yes

2. Has the statistical analysis been performed appropriately and rigorously?

Reviewer #1: Yes

Reviewer #2: Yes

3. Have the authors made all data underlying the findings in their manuscript fully available?

Reviewer #1: Yes

Reviewer #2: Yes

4. Is the manuscript presented in an intelligible fashion and written in standard English?

Reviewer #1: Yes

Reviewer #2: Yes

Reviewer #1: The manuscript is technically sound and the data support the conclusions. Limitations of the study has been addressed. Bias has been adequately assessed.

The manuscript is written in standard English.

The manuscript has all the data available.

Statistical analysis has been performed appropriately and rigorously.

Reviewer #2: rigorous and unnecessary detail.

suggest to highlight only the important findings.

abstract, good

Introduction, line 66-69 suggest to remove. just highlight the advantages of your nomogram without undermining other scoring systems. General statements are acceptable.

line 83 (back box nature?)

Don't include all table, just include the one with significant findings.

Discussion: Change to address limitations of existing tools.

eg, first validation tool that incorporates nomogram..

The differences with previous tool, eg RDW

**Do you want your identity to be public for this peer review?** For information about this choice, including consent withdrawal, please see our Privacy Policy

Reviewer #1: **Yes: ** Ankit Agarwal

Reviewer #2: **Yes: ** Huda Zainal Abidin

---

## [Author Response · Author response to Decision Letter 1]

30 May 2025

Response letter

Dear Editor and Reviewers,

Thank you for your letter and for the reviewers’ comments concerning our manuscript entitled“Development and temporal validation of a nomogram for predicting ICU 28-day mortality in middle-aged and elderly sepsis patients: An eICU database study

”(ID: PONE-D-25-14255). Those comments are all valuable and very helpful for revising and improving our paper, as well as the important guiding significance to our research. We have studied comments carefully and have made correction which we hope meet with approval. Revised portions are marked in red in the paper. The main corrections in the paper and the responses to the reviewer’s comments are as follows:

Responds to the reviewer’s comments:

Reviewer #1:

1. Response to comment: (The manuscript is technically sound and the data support the conclusions. Limitations of the study has been addressed. Bias has been adequately assessed. The manuscript is written in standard English. The manuscript has all the data available. Statistical analysis has been performed appropriately and rigorously.)

Response: We sincerely appreciate the reviewer’s thorough review and positive evaluation of our manuscript. We are grateful for your acknowledgment that our research is "technically sound and the data support the conclusions," and that "limitations of the study has been addressed, bias has been adequately assessed." We also thank you for recognizing that "the manuscript is written in standard English, has all the data available, and the statistical analysis has been performed appropriately and rigorously."

Your positive feedback is greatly encouraging to our research team. We have strived to ensure scientific rigor in our methodology and reliability in our results. Your professional recognition confirms that our efforts in these aspects have been successful.

Thank you again for your valuable time and expert opinion, which are important in enhancing the quality of our manuscript.

Reviewer #2:

1. Response to comment: (rigorous and unnecessary detail. suggest to highlight only the important findings)

Response: We greatly appreciate the reviewer's suggestion and have thoroughly revised the manuscript to streamline content and highlight key findings. Our modifications focused on three main areas: (1) in the Methods section, we simplified the description of statistical analyses; (2) in the Results section, we streamlined the explanations of tables and figures, focusing on primary outcomes; (3) in the Discussion section, we removed lengthy discussions of secondary predictive variables and reduced repetition of information already presented in the Results. Specific modifications for each section are detailed below:

Methods section:

1.Database extraction procedures and inclusion/exclusion criteria:

Lines 146-156: "This retrospective study utilized data from the eICU Collaborative Research Database (eICU-CRD), which contains detailed clinical information from 335 intensive care units across 208 hospitals in the United States between 2014 and 2015. We accessed the database for research purposes on March 18, 2025, after one of our researchers completed the CITI program training and obtained PhysioNet certification (Record ID: 67403327). The database is publicly available and completely deidentified, certified as HIPAA compliant (Privacert, Cambridge, MA; Certification No. 1031219-2). The researchers had no access to identifiable patient information either during the original data collection period or during our analysis. Due to the deidentified nature of the database, the requirement for informed consent was waived (details available at: https://eicu-crd.mit.edu/about/acknowledgments/)."

were simplified to "We utilized the eICU Collaborative Research Database (eICU-CRD), which contains clinical information from 335 intensive care units across 208 hospitals in the United States from 2014 to 2015. We accessed the database after completing the CITI program training and obtaining PhysioNet certification (Record ID: 67403327). The database is publicly available and fully de-identified in compliance with HIPAA regulations (Privacert, Cambridge, MA; Certification No. 1031219-2)."

Lines 166-174:"We initially identified 23,136 patients with a diagnosis of sepsis from eICU-CRD diagnosis records. The following exclusion criteria were applied: (1) repeated ICU admissions (only the first ICU admission for each patient was included, 3,610 excluded), (2) ICU stays less than 24 hours (4,254 excluded), (3) age less than 45 years (1,553 excluded), and (4) missing ICU outcome data (2 excluded). After applying these exclusion criteria, a final study cohort of 13,717 patients was included for analysis. The cohort was temporally divided into a training cohort (patients admitted in 2014, n=6,397) and a validation cohort (patients admitted in 2015, n=7,320)." were simplified to "According to Sepsis-3 criteria, we included patients diagnosed with sepsis upon ICU admission. After excluding duplicate admissions, ICU stays less than 24 hours, patients younger than 45 years, and those with missing ICU outcome data, 13,717 patients were finally included and chronologically divided into training (2014, n=6,397) and validation cohorts (2015, n=7,320)."

2.Statistical methods description:

Lines 208-222: "During the model development phase, we first performed univariate logistic regression analysis on all variables to assess their associations with 28-day ICU mortality in the training cohort (Table S1). Variables demonstrating statistical significance (p < 0.05) were subsequently evaluated for multicollinearity via variance inflation factor (VIF) testing. Variables with a VIF > 5 were excluded to mitigate multicollinearity (Table S2), resulting in the removal of several variables, including bicarbonate (VIF=17.0), base excess (VIF=18.2), ALT (VIF=9.9), PT (VIF=66.2), APACHE IV score (VIF=69.6), and acute physiology score III (VIF=73.0).

For the remaining variables, we employed random forest modelling to evaluate their relative importance and contribution to the prediction of 28-day ICU mortality (Table S3). The random forest model was built using 500 trees, with six split features selected from the 34 total features, and sampling was performed without replacement. In parallel, we conducted LASSO regression with 10-fold cross-validation for variable selection, applying the one-standard-error rule to determine the optimal lambda value (lambda.1 se = 0.005) (Figure S1).” were simplified to: "From univariate analysis, we identified variables significantly associated with 28-day ICU mortality (p<0.05), removed those with high multicollinearity (VIF>5), and further refined our selection through random forest importance ranking and LASSO regression with 10-fold cross-validation, ultimately identifying 11 key predictors for the final model (detailed in Tables S1-S3 and Figure S1)."

Lines 229-237, which describe detailed model evaluation methods "The model's performance was evaluated on the basis of several metrics, including the area under the receiver operating characteristic curve (AUC), sensitivity, specificity, positive predictive value (PPV), and negative predictive value (NPV). Calibration was assessed via calibration curves, which revealed strong concordance between the predicted probabilities and observed outcomes in both the development and validation cohorts, with the curves aligning closely to the diagonal line. Additionally, the clinical utility of the model was evaluated via decision curve analysis (DCA). The final model was visualized as a nomogram." were condensed to: "Model performance was assessed through area under the receiver operating characteristic curve (AUC), sensitivity, specificity, positive predictive value (PPV), and negative predictive value (NPV). Calibration was evaluated using calibration curves, and clinical utility was assessed via decision curve analysis (DCA). The final model was visualized as a nomogram."

Results Section:

1.Reduce detailed baseline characteristics description:

Lines 260-269: "The training and validation cohorts were well matched in terms of most demographic and clinical characteristics, including age (69.5±12.3 vs 69.6±12.0 years), BMI (29.0± 9.0 vs 28.9±8.8 kg/m2), and gender distribution (49.4% vs 48.3% male). Disease severity was comparable between the groups, as demonstrated by similar SOFA scores (median 4.0, IQR 2.0-6.0) and APACHE IV scores (71.9±24.9 vs 71.1±25.1). However, compared with the training cohort, the validation cohort demonstrated significantly higher respiratory rates, oxygen saturation, and lactate levels, along with a lower incidence of diabetes and a higher incidence of acute myocardial infarction (all P<0.001)." were condensed to: "The training and validation cohorts exhibited similar demographic and clinical characteristics, with comparable disease severity as measured by SOFA and APACHE IV scores. Only minor differences were observed in respiratory parameters, lactate levels, and certain comorbidities between the cohorts (Table 1)."

2.Remove some supplementary analyses that do not directly contribute to the main conclusions

Lines 317-328:"We compared the performance of our predictive model with that of the previously published SMRS scoring system [16], which was developed via the LASSO method, to select 13 predictive variables from 35 clinical features. The SMRS, which is specifically designed for ICU patients meeting the sepsis-3 criteria, has a total score range of 0--34 points. In our study population, our model demonstrated slightly better discriminatory power, with an AUC of 0.813 (95% CI: 0.750-0.877), than the SMRS model, which had an AUC of 0.789 (95% CI: 0.720-0.859) (Figure 2). At the respective optimal thresholds (-1.536 for our model and -1.544 for SMRS), our model exhibited marginally superior diagnostic performance metrics, including sensitivity (80.36% vs. 78.57%), specificity (68.29% vs. 67.32%), positive predictive value (40.91% vs. 39.64%), negative predictive value (92.72% vs. 92.00%), and overall accuracy (70.88% vs. 69.73%) (Table S5)." were revised to "Our model demonstrated slightly better discriminatory power (AUC: 0.813, 95% CI: 0.750-0.877) than the previously published SMRS scoring system (AUC: 0.789, 95% CI: 0.720-0.859) for predicting sepsis mortality in our study population (Figure 2)." We removed the reference to Table S5 and the detailed diagnostic performance metrics (sensitivity, specificity, positive predictive value, negative predictive value, and accuracy) because these details are not directly essential to the main conclusion.

3.Simplify detailed explanations of model performance metrics

Lines 333-340: "As shown in Figure 3 and Table 3, the model demonstrated good discriminative ability in the training cohort (n=420; AUC: 0.805, 95% CI: 0.749-0.862) and maintained stable performance in the validation cohort (n=491; AUC: 0.756, 95% CI: 0.700-0.812). The sensitivity (73.81% vs 70.59%), specificity (74.70% vs 71.47%), and predictive values (PPV: 42.18% vs 39.34%; NPV: 91.94% vs 90.26%) remained consistent between the training and validation cohorts. The calibration curves demonstrated good agreement between the predicted and observed probabilities in both cohorts (Figure 4)." were condensed to: "Our predictive model showed good discrimination in both the training cohort (AUC: 0.805) and validation cohort (AUC: 0.756), with consistent sensitivity, specificity, and predictive values across cohorts (Table 3, Figure 3). Calibration curves confirmed good agreement between predicted and observed mortality probabilities (Figure 4)."

Discussion Section:

1.Concise discussion of each individual predictor variable - focus only on the most novel or clinically significant variables:

Lines 414-421: "Our study identified 11 key predictors of sepsis mortality through a comprehensive analysis. These predictors cover multiple physiological systems and reflect different aspects of disease severity, including hemodynamics (heart rate), respiratory function (RDW), neurological status (GCS score), organ dysfunction (SOFA score), tissue perfusion (lactate, pH), renal function (24 h urine output), coagulation (platelet count), nutritional status (total protein), and systemic response (temperature, WBC count). The selection of these variables is consistent with previous studies and aligns with the pathophysiological mechanisms of sepsis." were condensed to: "Our study identified 11 key predictors of sepsis mortality, with RDW, SOFA score, and lactate level emerging as the most clinically significant. These three variables reflect critical aspects of sepsis pathophysiology including inflammatory status, organ dysfunction, and tissue hypoperfusion."

2.Reduce repetitive statements about model performance:

Lines 384-394: "Building upon these findings, we developed and temporally validated a novel nomogram incorporating 11 readily available clinical predictors. The model demonstrated superior discriminative ability (AUC: 0.813 vs 0.789) with robust performance in both the training (n=6,397, mortality: 9.08%, sensitivity: 73.81%, specificity: 74.70%) and validation (n=7,320, mortality: 9.49%, sensitivity: 70.59%, specificity: 71.47%) cohorts. Decision curve analysis further confirmed the clinical utility of our model, showing greater net benefit across a wide range of threshold probabilities than the treat-all/treat-none strategies did. This represents the first temporally validated nomogram specifically designed for middle-aged and elderly sepsis patients, providing an interpretable and practical tool for early risk stratification." were revised to "Our research introduces innovation to the field of sepsis prognostic assessment through the implementation of a nomogram as a visual tool. Compared to traditional prediction methods, nomograms intuitively present the weights of various risk factors, enabling clinicians to more clearly understand each factor's influence on prognosis. We developed and validated a novel nomogram incorporating 11 clinical predictors that outperforms existing scoring systems, representing the first temporally validated risk stratification tool specifically designed for middle-aged and elderly sepsis patients, addressing a gap in existing assessment approaches regarding visual presentation and clinical practicality."

Lines 462-470: "The nomogram incorporates 11 routine clinical parameters (RDW, SOFA score, lactate, pH, etc.), enabling clinicians to assess 28-day ICU mortality risk quickly in middle-aged and elderly sepsis patients. The model demonstrated robust performance in both the training (AUC: 0.805) and validation (AUC: 0.756) cohorts, outperforming conventional scoring systems (AUC: 0.813 vs 0.789), with decision curve analysis showing favourable clinical utility. This tool facilitates the early identification of high-risk patients and optimizes clinical decision-making and resource allocation, potentially improving outcomes for elderly sepsis patients in ICU settings." were condensed to: "Our nomogram provides clinicians with an intuitive tool to assess 28-day ICU mortality risk in middle-aged and elderly sepsis patients, potentially facilitating early identification of high-risk individuals and optimizing resource allocation in ICU settings." This not only clarifies the clinical significance of this study, but also avoids duplication with the results.

2.Response to comment: (Introduction, line 66-69 suggest to remove. just highlight the advantages of your nomogram without undermining other scoring systems. General statements are acceptable.)

Response: We appreciate the reviewer's constructive feedback regarding our introduction. We agree that it's more appropriate to focus on the advantages of our nomogram rather than emphasizing limitations of existing systems. Following the reviewer's suggestion, we have removed the specific criticisms of APACHE II and SAPS II scores (lines 66-69) and replaced them with the following more general statement:

"While existing scoring systems provide valuable guidance in clinical practice, our nomogram model aims to offer more individualized risk assessment for middle-aged and elderly sepsis patients by integrating multiple clinical and laboratory parameters

---

## [Decision Letter · Decision Letter 1]

Development and temporal validation of a nomogram for predicting ICU 28-day mortality in middle-aged and elderly sepsis patients: An eICU database study

PONE-D-25-14255R1

Dear Dr. Cui,

We’re pleased to inform you that your manuscript has been judged scientifically suitable for publication and will be formally accepted for publication once it meets all outstanding technical requirements.

Kind regards,

Robert Jeenchen Chen, MD, MPH, ChFC®, EA

Academic Editor

PLOS ONE

Additional Editor Comments (optional):

Reviewers' comments:

Reviewer's Responses to Questions

**Comments to the Author**

Reviewer #2: All comments have been addressed

Reviewer #3: All comments have been addressed

2. Is the manuscript technically sound, and do the data support the conclusions?

Reviewer #2: Yes

Reviewer #3: Yes

3. Has the statistical analysis been performed appropriately and rigorously?

Reviewer #2: Yes

Reviewer #3: Yes

4. Have the authors made all data underlying the findings in their manuscript fully available?

Reviewer #2: Yes

Reviewer #3: Yes

5. Is the manuscript presented in an intelligible fashion and written in standard English?

Reviewer #2: Yes

Reviewer #3: Yes

Reviewer #2: excellent, well written, informative study. Suggested corrections have been made accordingly. continuation further of the study may provide more concrete findings.

Reviewer #3: The revised version of the manuscript was interesting and good written and discussed.

All comments of reviewers were addressed in the manuscript.

**Do you want your identity to be public for this peer review?** For information about this choice, including consent withdrawal, please see our Privacy Policy

Reviewer #2: **Yes: ** Huda Zainal Abidin

Reviewer #3: No

---

## [Editor Report · Acceptance letter]

PONE-D-25-14255R1

PLOS ONE

Dear Dr. Cui,

I'm pleased to inform you that your manuscript has been deemed suitable for publication in PLOS ONE. Congratulations! Your manuscript is now being handed over to our production team.

Kind regards,

on behalf of

Dr. Robert Jeenchen Chen

Academic Editor

PLOS ONE